# Sequential Neural Models with Stochastic Layers

**Marco Fraccaro**[†]    **Søren Kaae Sønderby**[‡]    **Ulrich Paquet**[*]    **Ole Winther**[†‡]

† Technical University of Denmark
‡ University of Copenhagen
* Google DeepMind

## Abstract

How can we efficiently propagate uncertainty in a latent state representation with recurrent neural networks? This paper introduces *stochastic recurrent neural networks* which glue a deterministic recurrent neural network and a state space model together to form a stochastic and sequential neural generative model. The clear separation of deterministic and stochastic layers allows a structured variational inference network to track the factorization of the model's posterior distribution. By retaining both the nonlinear recursive structure of a recurrent neural network and averaging over the uncertainty in a latent path, like a state space model, we improve the state of the art results on the Blizzard and TIMIT speech modeling data sets by a large margin, while achieving comparable performances to competing methods on polyphonic music modeling.

## 1   Introduction

Recurrent neural networks (RNNs) are able to represent long-term dependencies in sequential data, by adapting and propagating a deterministic hidden (or latent) state [5, 16]. There is recent evidence that when complex sequences such as speech and music are modeled, the performances of RNNs can be dramatically improved when uncertainty is included in their hidden states [3, 4, 7, 11, 12, 15]. In this paper we add a new direction to the explorer's map of treating the hidden RNN states as uncertain paths, by including the world of state space models (SSMs) as an RNN layer. By cleanly delineating a SSM layer, certain independence properties of variables arise, which are beneficial for making efficient posterior inferences. The result is a generative model for sequential data, with a matching inference network that has its roots in variational auto-encoders (VAEs).

SSMs can be viewed as a probabilistic extension of RNNs, where the hidden states are assumed to be random variables. Although SSMs have an illustrious history [24], their stochasticity has limited their widespread use in the deep learning community, as inference can only be exact for two relatively simple classes of SSMs, namely hidden Markov models and linear Gaussian models, neither of which are well-suited to modeling long-term dependencies and complex probability distributions over high-dimensional sequences. On the other hand, modern RNNs rely on gated nonlinearities such as long short-term memory (LSTM) [16] cells or gated recurrent units (GRUs) [6], that let the deterministic hidden state of the RNN act as an internal memory for the model. This internal memory seems fundamental to capturing complex relationships in the data through a statistical model.

This paper introduces the *stochastic recurrent neural network (SRNN)* in Section 3. SRNNs combine the gated activation mechanism of RNNs with the stochastic states of SSMs, and are formed by stacking a RNN and a nonlinear SSM. The state transitions of the SSM are nonlinear and are parameterized by a neural network that also depends on the corresponding RNN hidden state. The SSM can therefore utilize long-term information captured by the RNN.

We use recent advances in variational inference to efficiently approximate the intractable posterior distribution over the latent states with an inference network [19, 23]. The form of our variational

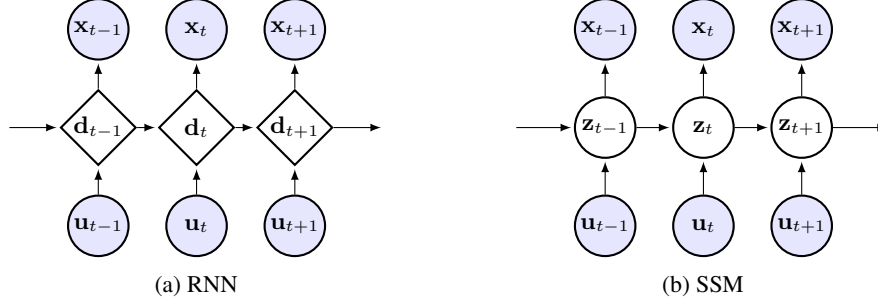

(a) RNN           (b) SSM

Figure 1: Graphical models to generate $\mathbf{x}_{1:T}$ with a recurrent neural network (RNN) and a state space model (SSM). Diamond-shaped units are used for deterministic states, while circles are used for stochastic ones. For sequence generation, like in a language model, one can use $\mathbf{u}_t = \mathbf{x}_{t-1}$.

approximation is inspired by the independence properties of the true posterior distribution over the latent states of the model, and allows us to improve inference by conveniently using the information coming from the whole sequence at each time step. The posterior distribution over the latent states of the SRNN is highly non-stationary while we are learning the parameters of the model. To further improve the variational approximation, we show that we can construct the inference network so that it only needs to learn how to compute the mean of the variational approximation at each time step given the mean of the *predictive* prior distribution.

In Section 4 we test the performances of SRNN on speech and polyphonic music modeling tasks. SRNN improves the state of the art results on the Blizzard and TIMIT speech data sets by a large margin, and performs comparably to competing models on polyphonic music modeling. Finally, other models that extend RNNs by adding stochastic units will be reviewed and compared to SRNN in Section 5.

## 2 Recurrent Neural Networks and State Space Models

*Recurrent neural networks* and *state space models* are widely used to model temporal sequences of vectors $\mathbf{x}_{1:T} = (\mathbf{x}_1, \mathbf{x}_2, \ldots, \mathbf{x}_T)$ that possibly depend on inputs $\mathbf{u}_{1:T} = (\mathbf{u}_1, \mathbf{u}_2, \ldots, \mathbf{u}_T)$. Both models rest on the assumption that the sequence $\mathbf{x}_{1:t}$ of observations up to time $t$ can be summarized by a latent state $\mathbf{d}_t$ or $\mathbf{z}_t$, which is deterministically determined ($\mathbf{d}_t$ in a RNN) or treated as a random variable which is averaged away ($\mathbf{z}_t$ in a SSM). The difference in treatment of the latent state has traditionally led to vastly different models: RNNs recursively compute $\mathbf{d}_t = f(\mathbf{d}_{t-1}, \mathbf{u}_t)$ using a parameterized nonlinear function $f$, like a LSTM cell or a GRU. The RNN observation probabilities $p(\mathbf{x}_t|\mathbf{d}_t)$ are equally modeled with nonlinear functions. SSMs, like linear Gaussian or hidden Markov models, explicitly model uncertainty in the latent process through $\mathbf{z}_{1:T}$. Parameter inference in a SSM requires $\mathbf{z}_{1:T}$ to be averaged out, and hence $p(\mathbf{z}_t|\mathbf{z}_{t-1}, \mathbf{u}_t)$ and $p(\mathbf{x}_t|\mathbf{z}_t)$ are often restricted to the exponential family of distributions to make many existing approximate inference algorithms applicable. On the other hand, averaging a function over the deterministic path $\mathbf{d}_{1:T}$ in a RNN is a trivial operation. The striking similarity in factorization between these models is illustrated in Figures 1a and 1b.

Can we combine the best of both worlds, and make the stochastic state transitions of SSMs nonlinear whilst keeping the gated activation mechanism of RNNs? Below, we show that a more expressive model can be created by stacking a SSM on top of a RNN, and that by keeping them layered, the functional form of the true posterior distribution over $\mathbf{z}_{1:T}$ guides the design of a backward-recursive structured variational approximation.

## 3 Stochastic Recurrent Neural Networks

We define a SRNN as a generative model $p_\theta$ by temporally interlocking a SSM with a RNN, as illustrated in Figure 2a. The joint probability of a single sequence and its latent states, assuming knowledge of the starting states $\mathbf{z}_0 = \mathbf{0}$ and $\mathbf{d}_0 = \mathbf{0}$, and inputs $\mathbf{u}_{1:T}$, factorizes as

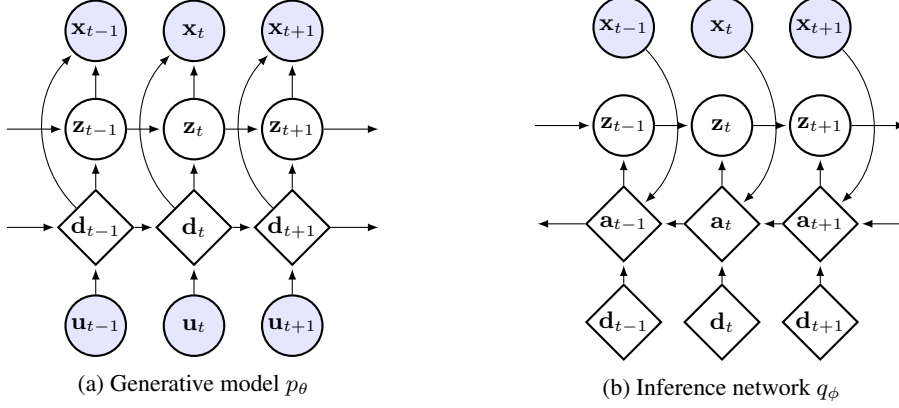

(a) Generative model $p_\theta$        (b) Inference network $q_\phi$

Figure 2: A SRNN as a generative model $p_\theta$ for a sequence $\mathbf{x}_{1:T}$. Posterior inference of $\mathbf{z}_{1:T}$ and $\mathbf{d}_{1:T}$ is done through an inference network $q_\phi$, which uses a backward-recurrent state $\mathbf{a}_t$ to approximate the nonlinear dependence of $\mathbf{z}_t$ on future observations $\mathbf{x}_{t:T}$ and states $\mathbf{d}_{t:T}$; see Equation (7).

$$p_\theta(\mathbf{x}_{1:T}, \mathbf{z}_{1:T}, \mathbf{d}_{1:T} | \mathbf{u}_{1:T}, \mathbf{z}_0, \mathbf{d}_0) = p_{\theta_\mathrm{x}}(\mathbf{x}_{1:T} | \mathbf{z}_{1:T}, \mathbf{d}_{1:T})\, p_{\theta_\mathrm{z}}(\mathbf{z}_{1:T} | \mathbf{d}_{1:T}, \mathbf{z}_0)\, p_{\theta_\mathrm{d}}(\mathbf{d}_{1:T} | \mathbf{u}_{1:T}, \mathbf{d}_0)$$

$$= \prod_{t=1}^{T} p_{\theta_\mathrm{x}}(\mathbf{x}_t | \mathbf{z}_t, \mathbf{d}_t)\, p_{\theta_\mathrm{z}}(\mathbf{z}_t | \mathbf{z}_{t-1}, \mathbf{d}_t)\, p_{\theta_\mathrm{d}}(\mathbf{d}_t | \mathbf{d}_{t-1}, \mathbf{u}_t)\,. \tag{1}$$

The SSM and RNN are further tied with skip-connections from $\mathbf{d}_t$ to $\mathbf{x}_t$. The joint density in (1) is parameterized by $\theta = \{\theta_\mathrm{x}, \theta_\mathrm{z}, \theta_\mathrm{d}\}$, which will be adapted together with parameters $\phi$ of a so-called "inference network" $q_\phi$ to best model $N$ independently observed data sequences $\{\mathbf{x}_{1:T_i}^i\}_{i=1}^N$ that are described by the log marginal likelihood or evidence

$$\mathcal{L}(\theta) = \log p_\theta\left(\{\mathbf{x}_{1:T_i}^i\} \mid \{\mathbf{u}_{1:T_i}^i, \mathbf{z}_0^i, \mathbf{d}_0^i\}_{i=1}^N\right) = \sum_i \log p_\theta(\mathbf{x}_{1:T_i}^i | \mathbf{u}_{1:T_i}^i, \mathbf{z}_0^i, \mathbf{d}_0^i) = \sum_i \mathcal{L}_i(\theta)\,. \tag{2}$$

Throughout the paper, we omit superscript $i$ when only one sequence is referred to, or when it is clear from the context. In each log likelihood term $\mathcal{L}_i(\theta)$ in (2), the latent states $\mathbf{z}_{1:T}$ and $\mathbf{d}_{1:T}$ were averaged out of (1). Integrating out $\mathbf{d}_{1:T}$ is done by simply substituting its deterministically obtained value, but $\mathbf{z}_{1:T}$ requires more care, and we return to it in Section 3.2. Following Figure 2a, the states $\mathbf{d}_{1:T}$ are determined from $\mathbf{d}_0$ and $\mathbf{u}_{1:T}$ through the recursion $\mathbf{d}_t = f_{\theta_\mathrm{d}}(\mathbf{d}_{t-1}, \mathbf{u}_t)$. In our implementation $f_{\theta_\mathrm{d}}$ is a GRU network with parameters $\theta_\mathrm{d}$. For later convenience we denote the value of $\mathbf{d}_{1:T}$, as computed by application of $f_{\theta_\mathrm{d}}$, by $\widetilde{\mathbf{d}}_{1:T}$. Therefore $p_{\theta_\mathrm{d}}(\mathbf{d}_t | \mathbf{d}_{t-1}, \mathbf{u}_t) = \delta(\mathbf{d}_t - \widetilde{\mathbf{d}}_t)$, i.e. $\mathbf{d}_{1:T}$ follows a delta distribution centered at $\widetilde{\mathbf{d}}_{1:T}$.

Unlike the VRNN [7], $\mathbf{z}_t$ directly depends on $\mathbf{z}_{t-1}$, as it does in a SSM, via $p_{\theta_\mathrm{z}}(\mathbf{z}_t | \mathbf{z}_{t-1}, \mathbf{d}_t)$. This split makes a clear separation between the deterministic and stochastic parts of $p_\theta$; the RNN remains entirely deterministic and its recurrent units do not depend on noisy samples of $\mathbf{z}_t$, while the prior over $\mathbf{z}_t$ follows the Markov structure of SSMs. The split allows us to later mimic the structure of the posterior distribution over $\mathbf{z}_{1:T}$ and $\mathbf{d}_{1:T}$ in its approximation $q_\phi$. We let the prior transition distribution $p_{\theta_\mathrm{z}}(\mathbf{z}_t | \mathbf{z}_{t-1}, \mathbf{d}_t) = \mathcal{N}(\mathbf{z}_t; \boldsymbol{\mu}_t^{(p)}, \mathbf{v}_t^{(p)})$ be a Gaussian with a diagonal covariance matrix, whose mean and log-variance are parameterized by neural networks that depend on $\mathbf{z}_{t-1}$ and $\mathbf{d}_t$,

$$\boldsymbol{\mu}_t^{(p)} = \mathrm{NN}_1^{(p)}(\mathbf{z}_{t-1}, \mathbf{d}_t)\,, \qquad\qquad \log \mathbf{v}_t^{(p)} = \mathrm{NN}_2^{(p)}(\mathbf{z}_{t-1}, \mathbf{d}_t)\,, \tag{3}$$

where NN denotes a neural network. Parameters $\theta_\mathrm{z}$ denote all weights of $\mathrm{NN}_1^{(p)}$ and $\mathrm{NN}_2^{(p)}$, which are two-layer feed-forward networks in our implementation. Similarly, the parameters of the emission distribution $p_{\theta_\mathrm{x}}(\mathbf{x}_t | \mathbf{z}_t, \mathbf{d}_t)$ depend on $\mathbf{z}_t$ and $\mathbf{d}_t$ through a similar neural network that is parameterized by $\theta_\mathrm{x}$.

## 3.1 Variational inference for the SRNN

The stochastic variables $\mathbf{z}_{1:T}$ of the nonlinear SSM cannot be analytically integrated out to obtain $\mathcal{L}(\theta)$ in (2). Instead of maximizing $\mathcal{L}$ with respect to $\theta$, we maximize a variational *evidence lower*

*bound* (ELBO) $\mathcal{F}(\theta, \phi) = \sum_i \mathcal{F}_i(\theta, \phi) \leq \mathcal{L}(\theta)$ with respect to both $\theta$ and the variational parameters $\phi$ [17]. The ELBO is a sum of lower bounds $\mathcal{F}_i(\theta, \phi) \leq \mathcal{L}_i(\theta)$, one for each sequence $i$,

$$\mathcal{F}_i(\theta, \phi) = \iint q_\phi(\mathbf{z}_{1:T}, \mathbf{d}_{1:T} | \mathbf{x}_{1:T}, A) \log \frac{p_\theta(\mathbf{x}_{1:T}, \mathbf{z}_{1:T}, \mathbf{d}_{1:T} | A)}{q_\phi(\mathbf{z}_{1:T}, \mathbf{d}_{1:T} | \mathbf{x}_{1:T}, A)} \, \mathrm{d}\mathbf{z}_{1:T} \, \mathrm{d}\mathbf{d}_{1:T} \,, \tag{4}$$

where $A = \{\mathbf{u}_{1:T}, \mathbf{z}_0, \mathbf{d}_0\}$ is a notational shorthand. Each sequence's approximation $q_\phi$ shares parameters $\phi$ with all others, to form the *auto-encoding variational Bayes* inference network or variational auto encoder (VAE) [19, 23] shown in Figure 2b. Maximizing $\mathcal{F}(\theta, \phi)$ – which we call "training" the neural network architecture with parameters $\theta$ and $\phi$ – is done by stochastic gradient ascent, and in doing so, both the posterior and its approximation $q_\phi$ change simultaneously. All the intractable expectations in (4) would typically be approximated by sampling, using the reparameterization trick [19, 23] or control variates [22] to obtain low-variance estimators of its gradients. We use the reparameterization trick in our implementation. Iteratively maximizing $\mathcal{F}$ over $\theta$ and $\phi$ separately would yield an expectation maximization-type algorithm, which has formed a backbone of statistical modeling for many decades [8]. The tightness of the bound depends on how well we can approximate the $i = 1, \ldots, N$ factors $p_\theta(\mathbf{z}_{1:T_i}^i, \mathbf{d}_{1:T_i}^i | \mathbf{x}_{1:T_i}^i, A^i)$ that constitute the true posterior over all latent variables with their corresponding factors $q_\phi(\mathbf{z}_{1:T_i}^i, \mathbf{d}_{1:T_i}^i | \mathbf{x}_{1:T_i}^i, A^i)$. In what follows, we show how $q_\phi$ could be judiciously structured to match the posterior factors.

We add initial structure to $q_\phi$ by noticing that the prior $p_{\theta_\mathrm{d}}(\mathbf{d}_{1:T} | \mathbf{u}_{1:T}, \mathbf{d}_0)$ in the generative model is a delta function over $\widetilde{\mathbf{d}}_{1:T}$, and so is the posterior $p_\theta(\mathbf{d}_{1:T} | \mathbf{x}_{1:T}, \mathbf{u}_{1:T}, \mathbf{d}_0)$. Consequently, we let the inference network use exactly the same deterministic state setting $\widetilde{\mathbf{d}}_{1:T}$ as that of the generative model, and we decompose it as

$$q_\phi(\mathbf{z}_{1:T}, \mathbf{d}_{1:T} | \mathbf{x}_{1:T}, \mathbf{u}_{1:T}, \mathbf{z}_0, \mathbf{d}_0) = q_\phi(\mathbf{z}_{1:T} | \mathbf{d}_{1:T}, \mathbf{x}_{1:T}, \mathbf{z}_0) \underbrace{q(\mathbf{d}_{1:T} | \mathbf{x}_{1:T}, \mathbf{u}_{1:T}, \mathbf{d}_0)}_{= \, p_{\theta_\mathrm{d}}(\mathbf{d}_{1:T} | \mathbf{u}_{1:T}, \mathbf{d}_0)} \,. \tag{5}$$

This choice exactly approximates one delta-function by itself, and simplifies the ELBO by letting them cancel out. By further taking the outer average in (4), one obtains

$$\mathcal{F}_i(\theta, \phi) = \mathbb{E}_{q_\phi} \left[ \log p_\theta(\mathbf{x}_{1:T} | \mathbf{z}_{1:T}, \widetilde{\mathbf{d}}_{1:T}) \right] - \mathrm{KL} \left( q_\phi(\mathbf{z}_{1:T} | \widetilde{\mathbf{d}}_{1:T}, \mathbf{x}_{1:T}, \mathbf{z}_0) \, \middle\| \, p_\theta(\mathbf{z}_{1:T} | \widetilde{\mathbf{d}}_{1:T}, \mathbf{z}_0) \right) \,, \tag{6}$$

which still depends on $\theta_\mathrm{d}$, $\mathbf{u}_{1:T}$ and $\mathbf{d}_0$ via $\widetilde{\mathbf{d}}_{1:T}$. The first term is an expected log likelihood under $q_\phi(\mathbf{z}_{1:T} | \widetilde{\mathbf{d}}_{1:T}, \mathbf{x}_{1:T}, \mathbf{z}_0)$, while KL denotes the Kullback-Leibler divergence between two distributions. Having stated the second factor in (5), we now turn our attention to parameterizing the first factor in (5) to resemble its posterior equivalent, by exploiting the temporal structure of $p_\theta$.

## 3.2 Exploiting the temporal structure

The true posterior distribution of the stochastic states $\mathbf{z}_{1:T}$, given both the data *and* the deterministic states $\mathbf{d}_{1:T}$, factorizes as $p_\theta(\mathbf{z}_{1:T} | \mathbf{d}_{1:T}, \mathbf{x}_{1:T}, \mathbf{u}_{1:T}, \mathbf{z}_0) = \prod_t p_\theta(\mathbf{z}_t | \mathbf{z}_{t-1}, \mathbf{d}_{t:T}, \mathbf{x}_{t:T})$. This can be verified by considering the conditional independence properties of the graphical model in Figure 2a using *d-separation* [13]. This shows that, knowing $\mathbf{z}_{t-1}$, the posterior distribution of $\mathbf{z}_t$ does not depend on the past outputs and deterministic states, but only on the present and future ones; this was also noted in [20]. Instead of factorizing $q_\phi$ as a mean-field approximation across time steps, we keep the structured form of the posterior factors, including $\mathbf{z}_t$'s dependence on $\mathbf{z}_{t-1}$, in the variational approximation

$$q_\phi(\mathbf{z}_{1:T} | \mathbf{d}_{1:T}, \mathbf{x}_{1:T}, \mathbf{z}_0) = \prod_t q_\phi(\mathbf{z}_t | \mathbf{z}_{t-1}, \mathbf{d}_{t:T}, \mathbf{x}_{t:T}) = \prod_t q_{\phi_\mathrm{z}}(\mathbf{z}_t | \mathbf{z}_{t-1}, \mathbf{a}_t = g_{\phi_\mathrm{a}}(\mathbf{a}_{t+1}, [\mathbf{d}_t, \mathbf{x}_t])), \tag{7}$$

where $[\mathbf{d}_t, \mathbf{x}_t]$ is the concatenation of the vectors $\mathbf{d}_t$ and $\mathbf{x}_t$. The graphical model for the inference network is shown in Figure 2b. Apart from the direct dependence of the posterior approximation at time $t$ on both $\mathbf{d}_{t:T}$ and $\mathbf{x}_{t:T}$, the distribution also depends on $\mathbf{d}_{1:t-1}$ and $\mathbf{x}_{1:t-1}$ through $\mathbf{z}_{t-1}$. We mimic each posterior factor's nonlinear long-term dependence on $\mathbf{d}_{t:T}$ and $\mathbf{x}_{t:T}$ through a backward-recurrent function $g_{\phi_\mathrm{a}}$, shown in (7), which we will return to in greater detail in Section 3.3. The inference network in Figure 2b is therefore parameterized by $\phi = \{\phi_\mathrm{z}, \phi_\mathrm{a}\}$ and $\theta_\mathrm{d}$.

In (7) all time steps are taken into account when constructing the variational approximation at time $t$; this can therefore be seen as a *smoothing* problem. In our experiments we also consider *filtering*,

where only the information up to time $t$ is used to define $q_\phi(\mathbf{z}_t|\mathbf{z}_{t-1}, \mathbf{d}_t, \mathbf{x}_t)$. As the parameters $\phi$ are shared across time steps, we can easily handle sequences of variable length in both cases.

As both the generative model and inference network factorize over time steps in (1) and (7), the ELBO in (6) separates as a sum over the time steps

$$\mathcal{F}_i(\theta, \phi) = \sum_t \mathbb{E}_{q_\phi^*(\mathbf{z}_{t-1})} \Big[ \mathbb{E}_{q_\phi(\mathbf{z}_t|\mathbf{z}_{t-1}, \widetilde{\mathbf{d}}_{t:T}, \mathbf{x}_{t:T})} \big[ \log p_\theta(\mathbf{x}_t|\mathbf{z}_t, \widetilde{\mathbf{d}}_t) \big] +$$
$$- \mathrm{KL}\Big( q_\phi(\mathbf{z}_t|\mathbf{z}_{t-1}, \widetilde{\mathbf{d}}_{t:T}, \mathbf{x}_{t:T}) \,\|\, p_\theta(\mathbf{z}_t|\mathbf{z}_{t-1}, \widetilde{\mathbf{d}}_t) \Big) \Big] , \qquad (8)$$

where $q_\phi^*(\mathbf{z}_{t-1})$ denotes the marginal distribution of $\mathbf{z}_{t-1}$ in the variational approximation to the posterior $q_\phi(\mathbf{z}_{1:t-1}|\widetilde{\mathbf{d}}_{1:T}, \mathbf{x}_{1:T}, \mathbf{z}_0)$, given by

$$q_\phi^*(\mathbf{z}_{t-1}) = \int q_\phi(\mathbf{z}_{1:t-1}|\widetilde{\mathbf{d}}_{1:T}, \mathbf{x}_{1:T}, \mathbf{z}_0) \, \mathrm{d}\mathbf{z}_{1:t-2} = \mathbb{E}_{q_\phi^*(\mathbf{z}_{t-2})} \Big[ q_\phi(\mathbf{z}_{t-1}|\mathbf{z}_{t-2}, \widetilde{\mathbf{d}}_{t-1:T}, \mathbf{x}_{t-1:T}) \Big] .$$
$$(9)$$

We can interpret (9) as having a VAE at each time step $t$, with the VAE being conditioned on the past through the stochastic variable $\mathbf{z}_{t-1}$. To compute (8), the dependence on $\mathbf{z}_{t-1}$ needs to be integrated out, using our posterior knowledge at time $t-1$ which is given by $q_\phi^*(\mathbf{z}_{t-1})$. We approximate the outer expectation in (8) using a Monte Carlo estimate, as samples from $q_\phi^*(\mathbf{z}_{t-1})$ can be efficiently obtained by ancestral sampling. The sequential formulation of the inference model in (7) allows such samples to be drawn and reused, as given a sample $\mathbf{z}_{t-2}^{(s)}$ from $q_\phi^*(\mathbf{z}_{t-2})$, a sample $\mathbf{z}_{t-1}^{(s)}$ from $q_\phi(\mathbf{z}_{t-1}|\mathbf{z}_{t-2}^{(s)}, \widetilde{\mathbf{d}}_{t-1:T}, \mathbf{x}_{t-1:T})$ will be distributed according to $q_\phi^*(\mathbf{z}_{t-1})$.

### 3.3 Parameterization of the inference network

The variational distribution $q_\phi(\mathbf{z}_t|\mathbf{z}_{t-1}, \mathbf{d}_{t:T}, \mathbf{x}_{t:T})$ needs to approximate the dependence of the true posterior $p_\theta(\mathbf{z}_t|\mathbf{z}_{t-1}, \mathbf{d}_{t:T}, \mathbf{x}_{t:T})$ on $\mathbf{d}_{t:T}$ and $\mathbf{x}_{t:T}$, and as alluded to in (7), this is done by running a RNN with inputs $\widetilde{\mathbf{d}}_{t:T}$ and $\mathbf{x}_{t:T}$ backwards in time. Specifically, we initialize the hidden state of the backward-recursive RNN in Figure 2b as $\mathbf{a}_{T+1} = \mathbf{0}$, and recursively compute $\mathbf{a}_t = g_{\phi_a}(\mathbf{a}_{t+1}, [\widetilde{\mathbf{d}}_t, \mathbf{x}_t])$. The function $g_{\phi_a}$ represents a recurrent neural network with, for example, LSTM or GRU units. Each sequence's variational approximation factorizes over time with $q_\phi(\mathbf{z}_{1:T}|\mathbf{d}_{1:T}, \mathbf{x}_{1:T}, \mathbf{z}_0) = \prod_t q_{\phi_z}(\mathbf{z}_t|\mathbf{z}_{t-1}, \mathbf{a}_t)$, as shown in (7). We let $q_{\phi_z}(\mathbf{z}_t|\mathbf{z}_{t-1}, \mathbf{a}_t)$ be a Gaussian with diagonal covariance, whose mean and the log-variance are parameterized with $\phi_z$ as

$$\boldsymbol{\mu}_t^{(q)} = \mathrm{NN}_1^{(q)}(\mathbf{z}_{t-1}, \mathbf{a}_t) , \qquad \log \mathbf{v}_t^{(q)} = \mathrm{NN}_2^{(q)}(\mathbf{z}_{t-1}, \mathbf{a}_t) . \qquad (10)$$

Instead of smoothing, we can also do filtering by using a neural network to approximate the dependence of the true posterior $p_\theta(\mathbf{z}_t|\mathbf{z}_{t-1}, \mathbf{d}_t, \mathbf{x}_t)$ on $\mathbf{d}_t$ and $\mathbf{x}_t$, through for instance $\mathbf{a}_t = \mathrm{NN}^{(a)}(\mathbf{d}_t, \mathbf{x}_t)$.

**Improving the posterior approximation.** In our experiments we found that during training, the parameterization introduced in (10) can lead to small values of the KL term $\mathrm{KL}(q_\phi(\mathbf{z}_t|\mathbf{z}_{t-1}, \mathbf{a}_t) \,\|\, p_\theta(\mathbf{z}_t|\mathbf{z}_{t-1}, \widetilde{\mathbf{d}}_t))$ in the ELBO in (8). This happens when $g_\phi$ in the inference network does not rely on the information propagated back from future outputs in $\mathbf{a}_t$, but it is mostly using the hidden state $\widetilde{\mathbf{d}}_t$ to imitate the behavior of the prior. The inference network could therefore get stuck by trying to optimize the ELBO through sampling from the prior of the model, making the variational approximation to the posterior useless. To overcome this issue, we directly include some knowledge of the *predictive* prior dynamics in the parameterization of the inference network, using our approximation of the posterior distribution $q_\phi^*(\mathbf{z}_{t-1})$ over the previous latent states. In the spirit of *sequential Monte Carlo* methods [10], we improve the parameterization of $q_\phi(\mathbf{z}_t|\mathbf{z}_{t-1}, \mathbf{a}_t)$ by using $q_\phi^*(\mathbf{z}_{t-1})$ from (9). As we are constructing the variational distribution sequentially, we approximate the predictive prior mean, i.e. our "best guess" on the prior dynamics of $\mathbf{z}_t$, as

$$\widehat{\boldsymbol{\mu}}_t^{(p)} = \int \mathrm{NN}_1^{(p)}(\mathbf{z}_{t-1}, \mathbf{d}_t) \, p(\mathbf{z}_{t-1}|\mathbf{x}_{1:T}) \, \mathrm{d}\mathbf{z}_{t-1} \approx \int \mathrm{NN}_1^{(p)}(\mathbf{z}_{t-1}, \mathbf{d}_t) \, q_\phi^*(\mathbf{z}_{t-1}) \, \mathrm{d}\mathbf{z}_{t-1} , \quad (11)$$

where we used the parameterization of the prior distribution in (3). We estimate the integral required to compute $\widehat{\boldsymbol{\mu}}_t^{(p)}$ by reusing the samples that were needed for the Monte Carlo estimate of the ELBO

in (8). This predictive prior mean can then be used in the parameterization of the mean of the variational approximation $q_\phi(\mathbf{z}_t|\mathbf{z}_{t-1}, \mathbf{a}_t)$,

$$\boldsymbol{\mu}_t^{(q)} = \widehat{\boldsymbol{\mu}}_t^{(p)} + \text{NN}_1^{(q)}(\mathbf{z}_{t-1}, \mathbf{a}_t) , \qquad (12)$$

and we refer to this parameterization as $\text{Res}_q$ in the results in Section 4. Rather than directly learning $\boldsymbol{\mu}_t^{(q)}$, we learn the *residual* between $\widehat{\boldsymbol{\mu}}_t^{(p)}$ and $\boldsymbol{\mu}_t^{(q)}$. It is straightforward to show that with this parameterization the KL-term in (8) will not depend on $\widehat{\boldsymbol{\mu}}_t^{(p)}$, but only on $\text{NN}_1^{(q)}(\mathbf{z}_{t-1}, \mathbf{a}_t)$. Learning the residual improves inference, making it seemingly easier for the inference network to track changes in the generative model while the model is trained, as it will only have to learn how to "correct" the predictive prior dynamics by using the information coming from $\widetilde{\mathbf{d}}_{t:T}$ and $\mathbf{x}_{t:T}$. We did not see any improvement in results by parameterizing $\log \mathbf{v}_t^{(q)}$ in a similar way. The inference procedure of SRNN with $\text{Res}_q$ parameterization for one sequence is summarized in Algorithm 1.

---

**Algorithm 1** Inference of SRNN with $\text{Res}_q$ parameterization from (12).

---
1: **inputs**: $\widetilde{\mathbf{d}}_{1:T}$ and $\mathbf{a}_{1:T}$
2: initialize $\mathbf{z}_0$
3: **for** $t = 1$ to $T$ **do**
4: $\quad \widehat{\boldsymbol{\mu}}_t^{(p)} = \text{NN}_1^{(p)}(\mathbf{z}_{t-1}, \widetilde{\mathbf{d}}_t)$
5: $\quad \boldsymbol{\mu}_t^{(q)} = \widehat{\boldsymbol{\mu}}_t^{(p)} + \text{NN}_1^{(q)}(\mathbf{z}_{t-1}, \mathbf{a}_t)$
6: $\quad \log \mathbf{v}_t^{(q)} = \text{NN}_2^{(q)}(\mathbf{z}_{t-1}, \mathbf{a}_t)$
7: $\quad \mathbf{z}_t \sim \mathcal{N}(\mathbf{z}_t; \boldsymbol{\mu}_t^{(q)}, \mathbf{v}_t^{(q)})$
8: **end for**

---

## 4 Results

In this section the SRNN is evaluated on the modeling of speech and polyphonic music data, as they have shown to be difficult to model without a good representation of the uncertainty in the latent states [3, 7, 11, 12, 15]. We test SRNN on the Blizzard [18] and TIMIT raw audio data sets (Table 1) used in [7]. The preprocessing of the data sets and the testing performance measures are identical to those reported in [7]. Blizzard is a dataset of 300 hours of English, spoken by a single female speaker. TIMIT is a dataset of 6300 English sentences read by 630 speakers. As done in [7], for Blizzard we report the average log-likelihood for half-second sequences and for TIMIT we report the average log likelihood per sequence for the test set sequences. Note that the sequences in the TIMIT test set are on average 3.1s long, and therefore 6 times longer than those in Blizzard. For the raw audio datasets we use a fully factorized Gaussian output distribution. Additionally, we test SRNN for modeling sequences of polyphonic music (Table 2), using the four data sets of MIDI songs introduced in [4]. Each data set contains more than 7 hours of polyphonic music of varying complexity: folk tunes (Nottingham data set), the four-part chorales by J. S. Bach (JSB chorales), orchestral music (MuseData) and classical piano music (Piano-midi.de). For polyphonic music we use a Bernoulli output distribution to model the binary sequences of piano notes. In our experiments we set $\mathbf{u}_t = \mathbf{x}_{t-1}$, but $\mathbf{u}_t$ could also be used to represent additional input information to the model.

All models where implemented using Theano [2], Lasagne [9] and Parmesan[1]. Training using a NVIDIA Titan X GPU took around 1.5 hours for TIMIT, 18 hours for Blizzard, less than 15 minutes for the JSB chorales and Piano-midi.de data sets, and around 30 minutes for the Nottingham and MuseData data sets. To reduce the computational requirements we use only 1 sample to approximate all the intractable expectations in the ELBO (notice that the KL term can be computed analytically). Further implementation and experimental details can be found in the Supplementary Material.

**Blizzard and TIMIT.** Table 1 compares the average log-likelihood per test sequence of SRNN to the results from [7]. For RNNs and VRNNs the authors of [7] test two different output distributions, namely a Gaussian distribution (Gauss) and a Gaussian Mixture Model (GMM). VRNN-I differs from the VRNN in that the prior over the latent variables is independent across time steps, and it is therefore similar to STORN [3]. For SRNN we compare the *smoothing* and *filtering* performance (denoted as *smooth* and *filt* in Table 1), both with the residual term from (12) and without it (10) (denoted as $\text{Res}_q$ if present). We prefer to only report the more conservative evidence lower bound for SRNN, as the approximation of the log-likelihood using standard importance sampling is known to be difficult to compute accurately in the sequential setting [10]. We see from Table 1 that SRNN outperforms all the competing methods for speech modeling. As the test sequences in TIMIT are on average more than 6 times longer than the ones for Blizzard, the results obtained with SRNN for

| Models | Blizzard | TIMIT |
|---|---|---|
| SRNN (smooth+Res$_q$) | $\geq$**11991** | $\geq$ **60550** |
| SRNN (smooth) | $\geq$ 10991 | $\geq$ 59269 |
| SRNN (filt+Res$_q$) | $\geq$ 10572 | $\geq$ 52126 |
| SRNN (filt) | $\geq$ 10846 | $\geq$ 50524 |
| VRNN-GMM | $\geq$ 9107 | $\geq$ 28982 |
| | $\approx$ 9392 | $\approx$ 29604 |
| VRNN-Gauss | $\geq$ 9223 | $\geq$ 28805 |
| | $\approx$ 9516 | $\approx$ 30235 |
| VRNN-I-Gauss | $\geq$ 8933 | $\geq$ 28340 |
| | $\approx$ 9188 | $\approx$ 29639 |
| RNN-GMM | 7413 | 26643 |
| RNN-Gauss | 3539 | -1900 |

Table 1: Average log-likelihood per sequence on the test sets. For TIMIT the average test set length is 3.1s, while the Blizzard sequences are all 0.5s long. The non-SRNN results are reported as in [7]. *Smooth*: $g_{\phi_a}$ is a GRU running backwards; *filt*: $g_{\phi_a}$ is a feed-forward network; Res$_q$: parameterization with residual in (12).

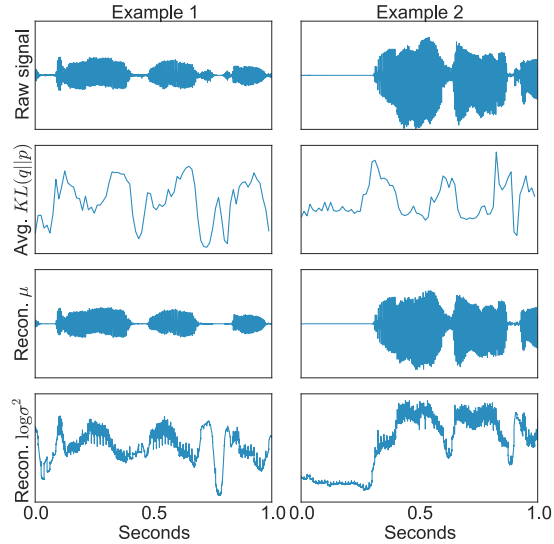

Figure 3: Visualization of the average KL term and reconstructions of the output mean and log-variance for two examples from the Blizzard test set.

| Models | Nottingham | JSB chorales | MuseData | Piano-midi.de |
|---|---|---|---|---|
| SRNN (smooth+Res$_q$) | $\geq -2.94$ | $\geq -4.74$ | $\geq -6.28$ | $\geq -8.20$ |
| TSBN | $\geq -3.67$ | $\geq -7.48$ | $\geq -6.81$ | $\geq -7.98$ |
| NASMC | $\approx -2.72$ | $\approx -\mathbf{3.99}$ | $\approx -6.89$ | $\approx -7.61$ |
| STORN | $\approx -2.85$ | $\approx -6.91$ | $\approx -6.16$ | $\approx -7.13$ |
| RNN-NADE | $\approx -\mathbf{2.31}$ | $\approx -5.19$ | $\approx -\mathbf{5.60}$ | $\approx -\mathbf{7.05}$ |
| RNN | $\approx -4.46$ | $\approx -8.71$ | $\approx -8.13$ | $\approx -8.37$ |

Table 2: Average log-likelihood on the test sets. The TSBN results are from [12], NASMC from [15], STORN from [3], RNN-NADE and RNN from [4].

TIMIT are in line with those obtained for Blizzard. The VRNN, which performs well when the voice of the single speaker from Blizzard is modeled, seems to encounter difficulties when modeling the 630 speakers in the TIMIT data set. As expected, for SRNN the variational approximation that is obtained when future information is also used (smoothing) is better than the one obtained by filtering. Learning the residual between the prior mean and the mean of the variational approximation, given in (12), further improves the performance in 3 out of 4 cases.

In the first two lines of Figure 3 we plot two raw signals from the Blizzard test set and the average KL term between the variational approximation and the prior distribution. We see that the KL term increases whenever there is a transition in the raw audio signal, meaning that the inference network is using the information coming from the output symbols to improve inference. Finally, the reconstructions of the output mean and log-variance in the last two lines of Figure 3 look consistent with the original signal.

**Polyphonic music.** Table 2 compares the average log-likelihood on the test sets obtained with SRNN and the models introduced in [3, 4, 12, 15]. As done for the speech data, we prefer to report the more conservative estimate of the ELBO in Table 2, rather than approximating the log-likelihood with importance sampling as some of the other methods do. We see that SRNN performs comparably to other state of the art methods in all four data sets. We report the results using smoothing and learning the residual between the mean of the predictive prior and the mean of the variational approximation, but the performances using filtering and directly learning the mean of the variational approximation are now similar. We believe that this is due to the small amount of data and the fact that modeling MIDI music is much simpler than modeling raw speech signals.

# 5   Related work

A number of works have extended RNNs with stochastic units to model motion capture, speech and music data [3, 7, 11, 12, 15]. The performances of these models are highly dependent on how the dependence among stochastic units is modeled over time, on the type of interaction between stochastic units and deterministic ones, and on the procedure that is used to evaluate the typically intractable log likelihood. Figure 4 highlights how SRNN differs from some of these works.

In STORN [3] *(Figure 4a)* and DRAW [14] the stochastic units at each time step have an isotropic Gaussian prior and are independent between time steps. The stochastic units are used as an input to the deterministic units in a RNN. As in our work, the reparameterization trick [19, 23] is used to optimize an ELBO.

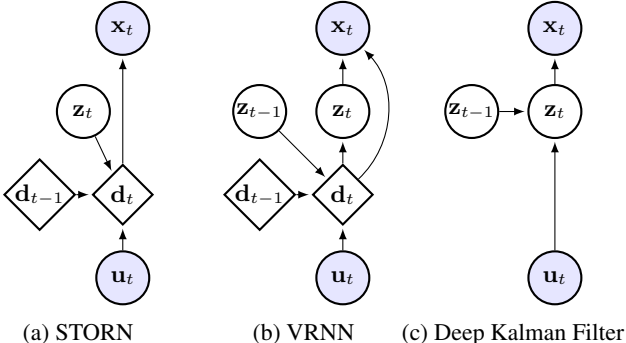

(a) STORN        (b) VRNN        (c) Deep Kalman Filter

Figure 4: Generative models of $\mathbf{x}_{1:T}$ that are related to SRNN. For sequence modeling it is typical to set $\mathbf{u}_t = \mathbf{x}_{t-1}$.

The authors of the VRNN [7] *(Figure 4b)* note that it is beneficial to add information coming from the past states to the prior over latent variables $\mathbf{z}_t$. The VRNN lets the prior $p_{\theta_z}(\mathbf{z}_t|\mathbf{d}_t)$ over the stochastic units depend on the deterministic units $\mathbf{d}_t$, which in turn depend on both the deterministic and the stochastic units at the previous time step through the recursion $\mathbf{d}_t = f(\mathbf{d}_{t-1}, \mathbf{z}_{t-1}, \mathbf{u}_t)$. The SRNN differs by clearly separating the deterministic and stochastic part, as shown in Figure 2a. The separation of deterministic and stochastic units allows us to improve the posterior approximation by doing smoothing, as the stochastic units still depend on each other when we condition on $\mathbf{d}_{1:T}$. In the VRNN, on the other hand, the stochastic units are conditionally independent given the states $\mathbf{d}_{1:T}$. Because the inference and generative networks in the VRNN share the deterministic units, the variational approximation would not improve by making it dependent on the future through $\mathbf{a}_t$, when calculated with a backward GRU, as we do in our model. Unlike STORN, DRAW and VRNN, the SRNN separates the "noisy" stochastic units from the deterministic ones, forming an entire layer of interconnected stochastic units. We found in practice that this gave better performance and was easier to train. The works by [1, 20] *(Figure 4c)* show that it is possible to improve inference in SSMs by using ideas from VAEs, similar to what is done in the stochastic part (the top layer) of SRNN. Towards the periphery of related works, [15] approximates the log likelihood of a SSM with sequential Monte Carlo, by learning flexible proposal distributions parameterized by deep networks, while [12] uses a recurrent model with discrete stochastic units that is optimized using the NVIL algorithm [21].

# 6   Conclusion

This work has shown how to extend the modeling capabilities of recurrent neural networks by combining them with nonlinear state space models. Inspired by the independence properties of the intractable true posterior distribution over the latent states, we designed an inference network in a principled way. The variational approximation for the stochastic layer was improved by using the information coming from the whole sequence and by using the $\mathrm{Res}_q$ parameterization to help the inference network to track the non-stationary posterior. SRNN achieves state of the art performances on the Blizzard and TIMIT speech data set, and performs comparably to competing methods for polyphonic music modeling.

# Acknowledgements

We thank Casper Kaae Sønderby and Lars Maaløe for many fruitful discussions, and NVIDIA Corporation for the donation of TITAN X and Tesla K40 GPUs. Marco Fraccaro is supported by Microsoft Research through its PhD Scholarship Programme.

## Footnotes

[1]github.com/casperkaae/parmesan. The code for SRNN is available at github.com/marcofraccaro/srnn.

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
