[Supplementary Material]

# Supplementary Material for Sequential Neural Models with Stochastic Layers

**Marco Fraccaro**[†]    **Søren Kaae Sønderby**[‡]    **Ulrich Paquet**[*]    **Ole Winther**[†‡]

[†] Technical University of Denmark
[‡] University of Copenhagen
[*] Google DeepMind

## 1    Experimental setup

### 1.1    Blizzard and TIMIT

The sampling rate is 16KHz and the raw audio signal is normalized using the global mean and standard deviation of the traning set. We split the raw audio signals in chunks of 2 seconds. The waveforms are then divided into non-overlapping vectors of size 200. The RNN thus runs for 160 steps[1]. The model is trained to predict the next vector ($\mathbf{x}_t$) given the current one ($\mathbf{u}_t$). During training we use backpropagation through time (BPTT) for 0.5 seconds, i.e we have 4 updates for each 2 seconds of audio. For the first 0.5 second we initialize hidden units with zeros and for the subsequent 3 chunks we use the previous hidden states as initialization.

For Blizzard we split the data using 90% for training, 5% for validation and 5% for testing. For testing we report the average log-likelihood per 0.5s sequences. For TIMIT we use the predefined test set for testing and split the rest of the data into 95% for training and 5% for validation. The training and testing setup are identical to the ones for Blizzard. For TIMIT the test sequences have variable length and are on average $3.1s$, i.e. more than 6 times longer than Blizzard.

We model the output using a fully factorized Gaussian distribution for $p_{\theta_x}(\mathbf{x}_t|\mathbf{z}_t, \mathbf{d}_t)$. The deterministic RNNs use GRUs [2], with 2048 units for Blizzard and 1024 units for TIMIT. In both cases, $\mathbf{z}_t$ is a 256-dimensional vector. All the neural networks have 2 layers, with 1024 units for Blizzard and 512 for TIMIT, and use leaky rectified nonlinearities with leakiness $\frac{1}{3}$ and clipped at $\pm 3$. In both generative and inference models we share a neural network to extract features from the raw audio signal. The sizes of the models were chosen to roughly match the number of parameters used in [3]. In all experiments it was fundamental to gradually introduce the KL term in the ELBO, as shown in [1, 6, 5]. We therefore multiply a temperature $\beta$ to the KL term, i.e. $\beta$KL, and linearly increase $\beta$ from 0.2 to 1 in the beginning of training (for Blizzard we increase it by 0.0001 after each update, while for TIMIT by 0.0003). In both data sets we used the ADAM optimizer [4]. For Blizzard we use a learning rate of 0.0003 and batch size of 128, for TIMIT they are 0.001 and 64 respectively.

### 1.2    Polyphonic music

We use the same model architecture as in the speech modeling experiments, except for the output Bernoulli variables used to model the active notes. We reduced the number of parameters in the model to 300 deterministic hidden units for the GRU networks, and 100 stochastic units whose distributions are parameterized with neural networks with 1 layer of 500 units.

## Footnotes

[1]2s·16Khz / 200 = 160