[Reviews · NeurIPS 2016]

Reviewer 1

Summary

In this paper the authors propose a model and method to train neural sequence models which contain a single layer of stochastic variables (per timestep). The model contains a RNN which is used to compute and propagate information forward in time and stochastic variables which are conditioned on these. A variational autoencoder architecture, which is optionally conditioned on a backward-running RNN, is used to perform approximate inference.

Qualitative Assessment

The paper is well written and explains how this work relates to previous work on variational autoencoding methods and to neural sequence models with random variables. The paragraph about improving the posterior approximation (165 ff.) hints at an interesting problem that probably deserves a more detailed analysis. Nevertheless, the authors propose a simple workaround that seems to work reasonably well (according to the results presented in table 3) and that even has a certain elegance to it. The empirical results are encouraging, especially on the relatively hard Blizzard and TIMIT datasets. They are less impressive on the polymorphic music datasets. Wouldn’t we expect that models with explicit stochastic variables work better / regularize better on these relatively small datasets?

Confidence in this Review

3-Expert (read the paper in detail, know the area, quite certain of my opinion)


Reviewer 2

Summary

This paper introduces a stochastic addition to recurrent neural networks. The paper is quite complex but reasonably well explained. Experiments are run on Blizzard and Timit datasets.

Qualitative Assessment

I would really like to see a RNN-CRF type baseline since there were several of those introduced, e.g. https://arxiv.org/abs/1603.00223 among others.

Confidence in this Review

1-Less confident (might not have understood significant parts)


Reviewer 3

Summary

The paper proposes a sequential generative model called the stochastic recurrent neural networks (SRNN). In SRNN, the deterministic RNN layer is connected to the state space model layer and this results in the step-wise factorization of the model's posterior distribution. In the experiments, significant improvements on the Blizzard and TIMIT dataset are reported while obtaining comparable results on the polyphonic musing modeling.

Qualitative Assessment

I really enjoyed the paper and the idea. It's written very clearly. And the experiment and its analysis are thorough. Also, the improvement is significant.

Confidence in this Review

2-Confident (read it all; understood it all reasonably well)


Reviewer 4

Summary

State space models (SSM) are widely used in statistics and machine learning to model time series data. The problem is that they cannot capture long-term dependencies. This paper provides a solution to this problem by parameterizing the state transitions of the SSM with a neural network that depends on the hidden state of a recurrent neural network - thus taking advantage of the memory of the rnn and introducing nonlinearity. The paper also presents a principled way of doing inference with this new model by building an inference network that captures the structure of the true posterior distribution of the latent states.

Qualitative Assessment

The paper is well written and adds significant improvement to previous work on adding stochasticity to recurrent neural nets. These improvements include: 1 - a new model for sequential data that can both capture long-term dependencies using recurrent neural nets and capture uncertainty well via stochastic latent states of state space models 2 - an inference network for doing posterior approximation with this new model...this inference method includes a solution to the vanishing KL term of the evidence lower bound... 3 - rich approximating variational distribution that encodes the same conditional independence relationship among the latent states as the true posterior distribution of these latent states. One avenue of improvement would be to include results of the VRNN model for the polyphonic music modeling experiment. The reason why this would be a great improvement is that the VRNN is the closest related work and the paper mentions the SRNN (new model as presented in the paper) yields better results because of the separation of the deterministic layer of the recurrent neural net and the stochastic layer of the state space model. Evidence of this is lacking for the second experiment.

Confidence in this Review

2-Confident (read it all; understood it all reasonably well)


Reviewer 5

Summary

The paper proposed a stochastic RNNs by stacking a deterministic layer with a stochastic layer in the recurrent architecture, where dependencies are imposed on sequential latent variables. To handle the dependency between the stochastic variables over time, a novel inference network based on GRUs is formulated to better approximate the posterior distribution. Experimental results are promising for speech and music data.

Qualitative Assessment

1. This is a natural extension of previously-considered stochastic recurrent models. The main benefit/motivation argued by the authors in Section 5 is that by adding dependencies between z_{t - 1} and z_t, the variational inference procedure can leverage a backward GRU to use future information in the sequence. However, as shown by the experimental results, the SRNN (filt) method already outperforms VRNN by a very large margin. Though adding GRUs can help, the improvement is relatively smaller. Thus the reason why SRNN is better than VRNN is a bit unclear at this point, since the dependency between z_{t - 1} and z_t can also be modeled indirectly through d_t in VRNN. It will be beneficial to add some theoretical analysis or experimental ablation study to resolve this puzzle. 2. It will be useful to report the results of different variants (e.g., w/o Req_q, filt) for the music data as well, so that one can analyze whether the effects of different techniques are similar across datasets (tasks). Overall, I think the model and the algorithm are novel and the experimental results are promising.

Confidence in this Review

2-Confident (read it all; understood it all reasonably well)


Reviewer 6

Summary

The authors combine state space models with RNNs. The RNNs form a stochastic layer while the state space model creates a stochastic layer. Their model is trained with the tractable variational evidence lower bound (ELBO). The model is evaluated on several speech and music datasets and also compare it to similar models like STORN, VRNN and Deep Kalman Filters.

Qualitative Assessment

Updates after Rebuttal: In the rebuttal the authors said (to clarify a concern of mine): "In the caption to Figure 1, we mentioned setting u_t=x_{t-1}, e.g. for language modelling. We used u_t=x_{t-1} for STORN and VRNN so that they could be compared more easily with the SRNN architecture in Figure 2. We will reiterate this in Figure’s 4 caption and elsewhere to keep it clear." I overlooked this information and it should indeed be mentioned in other captions and in the text. It is correct that when setting the action equal to the time-shifted observations all four models will have a more similar structure. However similar structure does not mean that the models are comparable (e.g. DKF still uses actions instead of observation as u_{1:T}). But this also means that there now is almost no difference between VRNN and the proposed method. The only difference is the missing connection from z to the deterministic hidden state. This does indeed "separate the noisy stochastic units from the deterministic ones", as the authors said, but this is not favorable when someone wants good generalization capabilities. After this clarification from the authors and the resulting changes in the model structure, I have to reduce the rating in the technical quality section and the novelty section. Initial review: The generative model for STORN in Figure 4(a) is wrong. STORN has no input from any action u_t and the deterministic hidden state d_t needs input from the last generated observation x_{t-1}. VRNN in Figure 4(b) is also missing a connection from the last generative sample x_{t-1} to the hidden state d_t. Technical quality: Experiments only evaluate negative log likelihoods and reconstruction quality. The generative model is never evaluated in itself, e.g., visualisations that show that the state transition is creating valid sequences even for multiple time-steps. Novelty/originality: The algorithm is different from STORN and VRNN but does not provide any novel improvements upon them. The experiments show that other algorithms perform even better on most datasets. Potential impact or usefulness: Impact is similar to STORN or VRNN since no clear improvements. Clarity and presentation: Quite clear.

Confidence in this Review

3-Expert (read the paper in detail, know the area, quite certain of my opinion)